# Hierarchical Autoregressive Modeling for Neural Video Compression

**Ruihan Yang**[1], **Yibo Yang**[1], **Joseph Marino**[2] and **Stephan Mandt**[1]
Department of Computer Science, UC Irvine[1]
Computation & Neural Systems, California Institute of Technology[2]
{ruihan.yang,yibo.yang,mandt}@uci.edu, jmarino@caltech.edu

## Abstract

Recent work by Marino et al. (2020) showed improved performance in sequential density estimation by combining masked autoregressive flows with hierarchical latent variable models. We draw a connection between such autoregressive generative models and the task of lossy video compression. Specifically, we view recent neural video compression methods (Lu et al., 2019; Yang et al., 2020b; Agustsson et al., 2020) as instances of a generalized stochastic temporal autoregressive transform, and propose avenues for enhancement based on this insight. Comprehensive evaluations on large-scale video data show improved rate-distortion performance over both state-of-the-art neural and conventional video compression methods.

## 1 Introduction

Recent advances in deep generative modeling have seen a surge of applications, including learning-based compression. Generative models have already demonstrated empirical improvements in image compression, outperforming classical codecs (Minnen et al., 2018; Yang et al., 2020d), such as BPG (Bellard, 2014). In contrast, the less developed area of neural video compression remains challenging due to complex temporal dependencies operating at multiple scales. Nevertheless, recent neural video codecs have shown promising performance gains (Agustsson et al., 2020), in some cases on par with current hand-designed, classical codecs, e.g., HEVC. Compared to hand-designed codecs, learnable codecs are not limited to specific data modality, and offer a promising approach for streaming specialized content, such as sports or video chats. Therefore, improving neural video compression is vital for dealing with the ever-growing amount of video content being created.

Source compression fundamentally involves decorrelation, i.e., transforming input data into white noise distributions that can be easily modeled and entropy-coded. Thus, improving a model's capability to decorrelate data automatically improves its compression performance. Likewise, we can improve the associated entropy model (i.e., the model's prior) to capture any remaining dependencies. Just as compression techniques attempt to *remove* structure, generative models attempt to *model* structure. One family of models, autoregressive flows, maps between less structured distributions, e.g., uncorrelated noise, and more structured distributions, e.g., images or video (Dinh et al., 2014; 2016). The inverse mapping can remove dependencies in the data, making it more amenable for compression. Thus, a natural question to ask is how autoregressive flows can best be utilized in compression, and if mechanisms in existing compression schemes can be interpreted as flows.

This paper draws on recent insights in hierarchical sequential latent variable models with autoregressive flows (Marino et al., 2020). In particular, we identify connections between this family of models and recent neural video codecs based on motion estimation (Lu et al., 2019; Agustsson et al., 2020). By interpreting this technique as an instantiation of a more general autoregressive flow transform, we propose various alternatives and improvements based on insights from generative modeling.

In more detail, our main contributions are as follows:

1. **A new framework.** We interpret existing video compression methods through the more general framework of generative modeling, variational inference, and autoregressive flows, allowing us to readily investigate extensions and ablations. In particular, we compare fully data-driven approaches with motion-estimation-based neural compression schemes, and

consider a more expressive prior model for better entropy coding (described in the second bullet point below). This framework also provides directions for future work.

2. **A new model.** Following the predictive coding paradigm of video compression (Wiegand et al., 2003), Scale-Space Flow (SSF)(Agustsson et al., 2020) uses motion estimation to predict the frame being compressed, and further compresses the residual obtained by subtraction. Our proposed model extends the SSF model with a more flexible decoder and prior, and improves over the state of the art in rate-distortion performance. Specifically, we

   • Incorporate a learnable scaling transform to allow for more expressive and accurate reconstruction. Augmenting a shift transform by scale-then-shift is inspired by improvements from extending NICE (Dinh et al., 2014) to RealNVP (Dinh et al., 2016).

   • Introduce a structured prior over the two sets of latent variables in the generative model of SSF, corresponding to jointly encoding the motion information and residual information. As the two tend to be spatially correlated, encoding residual information conditioned on motion information results in a more informed prior, and thus better entropy model, for the residual information; this cuts down the bit-rate for the latter that typically dominates the overall bit-rate.

3. **A new dataset.** The neural video compression community currently lacks large, high-resolution benchmark datasets. While we extensively experimented on the publicly available Vimeo-90k dataset (Xue et al., 2019), we also collected and utilized a larger dataset, YouTube-NT[1], available through executable scripts. Since no training data was publicly released for the previous state-of-the-art method (Agustsson et al., 2020), YouTube-NT would be a useful resource for making and comparing further progress in this field.

## 2 RELATED WORK

We divide related work into three categories: neural image compression, neural video compression, and sequential generative models.

**Neural Image Compression.** Considerable progress has been made by applying neural networks to image compression. Early works proposed by Toderici et al. (2017) and Johnston et al. (2018) leveraged LSTMs to model spatial correlations of the pixels within an image. Theis et al. (2017) first proposed an autoencoder architecture for image compression and used the straight-through estimator (Bengio et al., 2013) for learning a discrete latent representation. The connection to *probabilistic* generative models was drawn by Ballé et al. (2017), who firstly applied variational autoencoders (VAEs) (Kingma & Welling, 2013) to image compression. In subsequent work, Ballé et al. (2018) encoded images with a two-level VAE architecture involving a scale hyper-prior, which can be further improved by autoregressive structures (Minnen et al., 2018; Minnen & Singh, 2020) or by optimization at encoding time (Yang et al., 2020d). Yang et al. (2020e) and Flamich et al. (2019) demonstrated competitive image compression performance without a pre-defined quantization grid.

**Neural Video Compression.** Compared to image compression, video compression is a significantly more challenging problem, as statistical redundancies exist not only within each video frame (exploited by intra-frame compression) but also along the temporal dimension. Early works by Wu et al. (2018); Djelouah et al. (2019) and Han et al. (2019) performed video compression by predicting future frames using a recurrent neural network, whereas Chen et al. (2019) and Chen et al. (2017) used convolutional architectures within a traditional block-based motion estimation approach. These early approaches did not outperform the traditional H.264 codec and barely surpassed the MPEG-2 codec. Lu et al. (2019) adopted a hybrid architecture that combined a pre-trained Flownet (Dosovitskiy et al., 2015) and residual compression, which leads to an elaborate training scheme. Habibian et al. (2019) and Liu et al. (2020) combined 3D convolutions for dimensionality reduction with expressive autoregressive priors for better entropy modeling at the expense of parallelism and runtime efficiency. Our method extends a low-latency model proposed by Agustsson et al. (2020), which allows for end-to-end training, efficient online encoding and decoding, and parallelism.

---

[1]`https://github.com/privateyoung/Youtube-NT`

**Sequential Deep Generative Models.** We drew inspiration from a body of work on sequential generative modeling. Early deep learning architectures for dynamics forecasting involved RNNs (Chung et al., 2015). Denton & Fergus (2018) and Babaeizadeh et al. (2018) used VAE-based stochastic models in conjunction with LSTMs to model dynamics. Li & Mandt (2018) introduced both local and global latent variables for learning disentangled representations in videos. Other video generation models used generative adversarial networks (GANs) (Vondrick et al., 2016; Lee et al., 2018) or autoregressive models and normalizing flows (Rezende & Mohamed, 2015; Dinh et al., 2014; 2016; Kingma & Dhariwal, 2018; Kingma et al., 2016; Papamakarios et al., 2017). Recently, Marino et al. (2020) proposed to combine latent variable models with autoregressive flows for modeling dynamics at different levels of abstraction, which inspired our models and viewpoints.

## 3 VIDEO COMPRESSION THROUGH DEEP AUTOREGRESSIVE MODELING

We identify commonalities between hierarchical autoregressive flow models (Marino et al., 2020) and state-of-the-art neural video compression architectures (Agustsson et al., 2020), and will use this viewpoint to propose improvements on existing models.

### 3.1 BACKGROUND

We first review VAE-based compression schemes (Ballé et al., 2017) and formulate existing low-latency video codecs in this framework; we then review the related autoregressive flow model.

**Generative Modeling and Source Compression.** Given a a sequence of video frames $\mathbf{x}_{1:T}$, lossy compression seeks a compact description of $\mathbf{x}_{1:T}$ that simultaneously minimizes the description length $\mathcal{R}$ and information loss $\mathcal{D}$. The distortion $\mathcal{D}$ measures the reconstruction error caused by encoding $\mathbf{x}_{1:T}$ into a lossy representation $\bar{\mathbf{z}}_{1:T}$ and subsequently decoding it back to $\hat{\mathbf{x}}_{1:T}$, while $\mathcal{R}$ measures the bit rate (file size). In learned compression methods (Ballé et al., 2017; Theis et al., 2017), the above process is parameterized by flexible functions $f$ ("encoder") and $g$ ("decoder") that map between the video and its latent representation $\bar{\mathbf{z}}_{1:T} = f(\mathbf{x}_{1:T})$. The goal is to minimize a rate-distortion loss, with the tradeoff between the two controlled by a hyperparameter $\beta > 0$:

$$\mathcal{L} = \mathcal{D}(\mathbf{x}_{1:T}, g(\lfloor\bar{\mathbf{z}}_{1:T}\rceil)) + \beta\mathcal{R}(\lfloor\bar{\mathbf{z}}_{1:T}\rceil).$$

We adopt the end-to-end compression approach of Ballé et al. (2017), which approximates the rounding operations $\lfloor\cdot\rceil$ (required for entropy coding) by uniform noise injection to enable gradient-based optimization during training. With an appropriate choice of probability model $p(\mathbf{z}_{1:T})$, the relaxed version of above R-D (rate-distortion) objective then corresponds to the VAE objective:

$$\tilde{\mathcal{L}} = \mathbb{E}_{q(\mathbf{z}_{1:T}|\mathbf{x}_{1:T})}[-\log p(\mathbf{x}_{1:T}|\mathbf{z}_{1:T}) - \log p(\mathbf{z}_{1:T})]. \tag{1}$$

In this model, the likelihood $p(\mathbf{x}_{1:T}|\mathbf{z}_{1:T})$ follows a Gaussian distribution with mean $\hat{\mathbf{x}}_{1:T} = g(\mathbf{z}_{1:T})$ and diagonal covariance $\frac{\beta}{2\log 2}\mathbf{I}$, and the approximate posterior $q$ is chosen to be a unit-width uniform distribution (thus has zero differential entropy) whose mean $\bar{\mathbf{z}}_{1:T}$ is predicted by an amortized inference network $f$. The prior density $p(\mathbf{z}_{1:T})$ interpolates its discretized version, so that it measures the code length of discretized $\bar{\mathbf{z}}_{1:T}$ after entropy-coding.

**Low-Latency Sequential Compression** We specialize Eq. 1 to make it suitable for low-latency video compression, widely used in both conventional and recent neural codecs (Rippel et al., 2019; Agustsson et al., 2020). To this end, we encode and decode individual frames $\mathbf{x}_t$ in sequence. Given the ground truth current frame $\mathbf{x}_t$ and the previously reconstructed frames $\hat{\mathbf{x}}_{<t}$, the encoder is restricted to be of the form $\bar{\mathbf{z}}_t = f(\mathbf{x}_t, \hat{\mathbf{x}}_{<t})$, and similarly the decoder computes reconstruction sequentially based on previous reconstructions and the current encoding, $\hat{\mathbf{x}}_t = g(\hat{\mathbf{x}}_{<t}, \lfloor\bar{\mathbf{z}}_t\rceil)$. Existing codecs usually condition on a *single* reconstructed frame, substituting $\hat{\mathbf{x}}_{<t}$ by $\hat{\mathbf{x}}_{t-1}$ in favor of efficiency. In the language of variational inference, the sequential encoder corresponds to a variational posterior of the form $q(\mathbf{z}_t|\mathbf{x}_t, \mathbf{z}_{<t})$, i.e., filtering, and the sequential decoder corresponds to the likelihood $p(\mathbf{x}_t|\mathbf{z}_{\leq t}) = \mathcal{N}(\hat{\mathbf{x}}_t, \frac{\beta}{2\log 2}\mathbf{I})$; in both distributions, the probabilistic conditioning on $\mathbf{z}_{<t}$ is based on the observation that $\hat{\mathbf{x}}_{t-1}$ is a deterministic function of $\mathbf{z}_{<t}$, if we identify $\lfloor\bar{\mathbf{z}}_t\rceil$ with the random variable $\mathbf{z}_t$ and unroll the recurrence $\hat{\mathbf{x}}_t = g(\hat{\mathbf{x}}_{<t}, \mathbf{z}_t)$. As we show, all sequential compression approaches considered in this work follow this paradigm, and the form of the reconstruction transform $\hat{\mathbf{x}}$ determines the lowest hierarchy of the corresponding generative process of video $\mathbf{x}$.

**Masked Autoregressive Flow (MAF).** As a final component in neural sequence modeling, we discuss MAF (Papamakarios et al., 2017), which models the joint distribution of a sequence $p(\mathbf{x}_{1:T})$ in terms of a simpler distribution of its underlying noise variables $\mathbf{y}_{1:T}$ through the following autoregressive transform and its inverse:

$$\mathbf{x}_t = h_\mu(\mathbf{x}_{<t}) + h_\sigma(\mathbf{x}_{<t}) \odot \mathbf{y}_t; \; \Leftrightarrow \; \mathbf{y}_t = \frac{\mathbf{x}_t - h_\mu(\mathbf{x}_{<t})}{h_\sigma(\mathbf{x}_{<t})}. \tag{2}$$

The noise variable $\mathbf{y}_t$ usually comes from a standard normal distribution. While the forward MAF transforms a sequence of standard normal noises into a data sequence, the inverse flow "whitens" the data sequence and removes temporal correlations. Due to its invertible nature, MAF allows for exact likelihood computations, but as we will explain in Section 3.3, we will not exploit this aspect in compression but rather draw on its expressiveness in modeling conditional likelihoods.

## 3.2 A GENERAL FRAMEWORK FOR GENERATIVE VIDEO CODING

We now describe a general framework that captures several existing low-latency neural compression methods as specific instances and gives rise to the exploration of new models. To this end, we combine latent variable models with autoregressive flows into a joint framework. We consider a sequential decoding procedure of the following form:

$$\hat{\mathbf{x}}_t = h_\mu(\hat{\mathbf{x}}_{t-1}, \mathbf{w}_t) + h_\sigma(\hat{\mathbf{x}}_{t-1}, \mathbf{w}_t) \odot g_v(\mathbf{v}_t, \mathbf{w}_t). \tag{3}$$

Eq. 3 resembles the definition of the MAF in Eq. 2, but augments this transform with two sets of latent variables $\mathbf{w}_t, \mathbf{v}_t \sim p(\mathbf{w}_t, \mathbf{v}_t)$. Above, $h_\mu$ and $h_\sigma$ are functions that transform the previous reconstructed data frame $\hat{\mathbf{x}}_{t-1}$ along with $\mathbf{w}_t$ into a shift and scale parameter, respectively. The function $g_v(\mathbf{v}_t, \mathbf{w}_t)$ converts these latent variables into a noise variable that encodes residuals with respect to the mean next-frame prediction $h_\mu(\hat{\mathbf{x}}_{t-1}, \mathbf{w}_t)$.

This stochastic decoder model has several advantages over existing generative models for compression, such as simpler flows or sequential VAEs. First, the stochastic autoregressive transform $h_\mu(\hat{\mathbf{x}}_{t-1}, \mathbf{w}_t)$ involves a latent variable $\mathbf{w}_t$ and is therefore more expressive than a deterministic transform (Schmidt & Hofmann, 2018; Schmidt et al., 2019). Second, compared to MAF, the additional nonlinear transform $g_v(\mathbf{v}_t, \mathbf{w}_t)$ enables more expressive residual noise, reducing the burden on the entropy model. Finally, as visualized in Figure 2, the scale parameter $h_\sigma(\hat{\mathbf{x}}_{t-1}, \mathbf{w}_t)$ effectively acts as a gating mechanism, determining how much variance is explained in terms of the autoregressive transform and the residual noise distribution. This provides an added degree of flexibility, in a similar fashion to how RealNVP improves over NICE (Dinh et al., 2014; 2016).

Our approach is inspired by Marino et al. (2020) who analyzed a restricted version of the model in Eq. 3, aiming to hybridize autoregressive flows and sequential latent variable models for video prediction. In contrast to Eq. 3, their model involved deterministic transforms as well as residual noise that came from a sequential VAE.

## 3.3 EXAMPLE MODELS AND EXTENSIONS

Next, we will show that the general framework expressed by Eq. 3 captures a variety of state-of-the-art neural video compression schemes and gives rise to extensions and new models.

**Temporal Autoregressive Transform (TAT).** The first special case among the class of models that are captured by Eq. 3 is the autoregressive neural video compression model by Yang et al. (2020b), which we refer to as temporal autoregressive transform (TAT). Shown in Figure 1(a), the decoder $g$ implements a deterministic scale-shift autoregressive transform of decoded noise $\mathbf{y}_t$,

$$\hat{\mathbf{x}}_t = g(\mathbf{z}_t, \hat{\mathbf{x}}_{t-1}) = h_\mu(\hat{\mathbf{x}}_{t-1}) + h_\sigma(\hat{\mathbf{x}}_{t-1}) \odot \mathbf{y}_t, \quad \mathbf{y}_t = g_z(\mathbf{z}_t). \tag{4}$$

The encoder $f$ inverts the transform to decorrelate the input frame $\mathbf{x}_t$ into $\bar{\mathbf{y}}_t$ and encodes the result as $\bar{\mathbf{z}}_t = f(\mathbf{x}_t, \hat{\mathbf{x}}_{t-1}) = f_z(\bar{\mathbf{y}}_t)$, where $\bar{\mathbf{y}}_t = \frac{\mathbf{x}_t - h_\mu(\hat{\mathbf{x}}_{t-1})}{h_\sigma(\hat{\mathbf{x}}_{t-1})}$. The shift $h_\mu$ and scale $h_\sigma$ transforms are parameterized by neural networks, $f_z$ is a convolutional neural network (CNN), and $g_z$ is a deconvolutional neural network (DNN) that approximately inverts $f_z$.

The TAT decoder is a simple version of the more general stochastic autoregressive transform in Eq 3, where $h_\mu$ and $h_\sigma$ lack latent variables. Indeed, interpreting the probabilistic generative process of

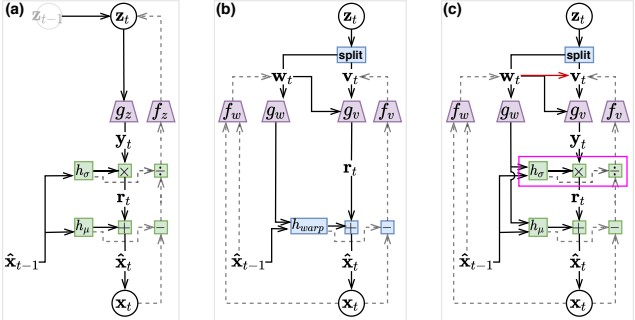

Figure 1: **Model diagrams** for the generative and inference procedures of the current frame $\mathbf{x}_t$, for various neural video compression methods. Random variables are shown in circles, all other quantities are deterministically computed; solid and dashed arrows describe computational dependency during generation (decoding) and inference (encoding), respectively. Purple nodes correspond to neural encoders (CNNs) and decoders (DNNs), and green nodes implement temporal autoregressive transform. (a) TAT; (b) SSF; (c) STAT or STAT-SSF; the magenta box highlights the additional proposed scale transform absent in SSF, and the red arrow from $\mathbf{w}_t$ to $\mathbf{v}_t$ highlights the proposed (optional) structured prior. See Appendix Fig. 7 for computational diagrams of the structured prior.

$\hat{\mathbf{x}}$, TAT implements the model proposed by Marino et al. (2020), as the transform from $\mathbf{y}$ to $\hat{\mathbf{x}}$ is a MAF. However, the generative process corresponding to compression (reviewed in Section 3.1) adds additional white noise to $\hat{\mathbf{x}}$, with $\mathbf{x} := \hat{\mathbf{x}} + \boldsymbol{\epsilon}, \boldsymbol{\epsilon} \sim \mathcal{N}(\mathbf{0}, \frac{\beta}{2\log 2}\mathbf{I})$. Thus, the generative process from $\mathbf{y}$ to $\mathbf{x}$ is no longer an autoregressive flow. Regardless, TAT was shown to better capture the low-level dynamics of video frames than the autoencoder $(f_z, g_z)$ alone, and the inverse transform decorrelates raw video frames to simplify the input to the encoder $f_z$ (Yang et al., 2020b).

**DVC (Lu et al., 2019) and Scale-Space Flow (SSF, Agustsson et al. (2020)).** The second class of models captured by Eq. 3 belong to the conventional video compression framework based on predictive coding (Cutler, 1952; Wiegand et al., 2003; Sullivan et al., 2012); both models make use of two sets of latent variables $\mathbf{z}_{1:T} = \{\mathbf{w}_{1:T}, \mathbf{v}_{1:T}\}$ to capture different aspects of information being compressed, where $\mathbf{w}$ captures estimated motion information used in warping prediction, and $\mathbf{v}$ helps capture residual error not predicted by warping.

Like most classical approaches to video compression by predictive coding, the reconstruction transform in the above models has the form of a prediction shifted by residual error (decoded noise), and lacks the scaling factor $h_\sigma$ compared to the autoregressive transform in Eq. 3

$$\hat{\mathbf{x}}_t = h_{warp}(\hat{\mathbf{x}}_{t-1}, g_w(\mathbf{w}_t)) + g_v(\mathbf{v}_t, \mathbf{w}_t). \tag{5}$$

Above, $g_w$ and $g_v$ are DNNs, $\mathbf{o}_t := g_w(\mathbf{w}_t)$ has the interpretation of an estimated optical flow (motion) field, $h_{warp}$ is the computer vision technique of warping, and the residual $\mathbf{r}_t := g_v(\mathbf{v}_t, \mathbf{w}_t) = \hat{\mathbf{x}}_t - h_{warp}(\hat{\mathbf{x}}_{t-1}, \mathbf{o}_t)$ represents the prediction error unaccounted for by warping. Lu et al. (2019) only makes use of $\mathbf{v}_t$ in the residual decoder $g_v$, and performs simple 2D warping by bi-linear interpretation; SSF (Agustsson et al., 2020) augments the optical flow (motion) field $\mathbf{o}_t$ with an additional scale field, and applies scale-space-warping to the progressively blurred versions of $\hat{\mathbf{x}}_{t-1}$ to allow for uncertainty in the warping prediction. The encoding procedure in the above models compute the variational mean parameters as $\bar{\mathbf{w}}_t = f_w(\mathbf{x}_t, \hat{\mathbf{x}}_{t-1}), \bar{\mathbf{v}}_t = f_v(\mathbf{x}_t - h_{warp}(\hat{\mathbf{x}}_{t-1}, g_w(\mathbf{w}_t)))$, corresponding to a structured posterior $q(\mathbf{z}_t|\mathbf{x}_t, \mathbf{z}_{<t}) = q(\mathbf{w}_t|\mathbf{x}_t, \mathbf{z}_{<t})q(\mathbf{v}_t|\mathbf{w}_t, \mathbf{x}_t, \mathbf{z}_{<t})$. We illustrate the above generative and inference procedures in Figure 1(b).

**Proposed: models based on Stochastic Temporal Autoregressive Transform.** Finally, we consider the most general models as described by the stochastic autoregressive transform in Eq. 3, shown in Figure 1(c). We study two main variants, categorized by how they implement $h_\mu$ and $h_\sigma$:

**STAT** uses DNNs for $h_\mu$ and $h_\sigma$ as in (Yang et al., 2020b), but complements it with the latent variable $\mathbf{w}_t$ that characterizes the transform. In principle, more flexible transforms should give better compression performance, but we find the following variant more parameter efficient in practice:
**STAT-SSF**: a less data-driven variant of the above that still uses scale-space warping (Agustsson

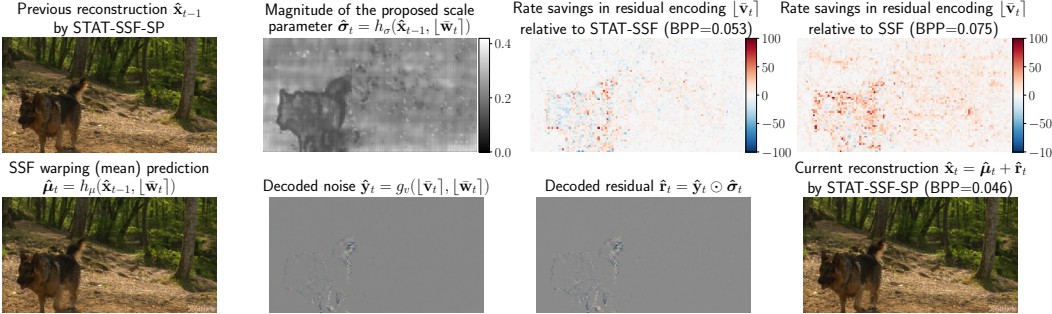

Figure 2: **Visualizing the proposed STAT-SSF-SP model** on one frame of UVG video "Shake-NDry". Two methods in comparison, STAT-SSF (proposed) and SSF (Agustsson et al., 2020), have comparable reconstruction quality to STAT-SSF-SP but higher bit-rate; the (BPP, PSNR) for STAT-SSF-SP, STAT-SSF, and SSF are (0.046, 36.97), (0.053, 36.94), and (0.075, 36.97), respectively. In this example, the warping prediction $\hat{\mu}_t = h_\mu(\hat{x}_{t-1}, \lfloor \bar{w}_t \rceil)$ incurs large error around the dog's moving contour, but models the mostly static background well, with the residual latents $\lfloor \bar{v}_t \rceil$ taking up an order of magnitude higher bit-rate than $\lfloor \bar{w}_t \rceil$ in the three methods. The proposed scale parameter $\hat{\sigma}_t$ gives the model extra flexibility when combining the noise $\hat{y}_t$ (decoded from $(\lfloor \bar{v}_t \rceil, \lfloor \bar{w}_t \rceil)$) with the warping prediction $\hat{\mu}_t$ (decoded from $\lfloor \bar{w}_t \rceil$ only) to form the reconstruction $\hat{x}_t = \hat{\mu}_t + \hat{\sigma}_t \odot \hat{y}_t$: the scale $\hat{\sigma}_t$ downweights contribution from the noise $\hat{y}_t$ in the foreground where it is very costly, and reduces the residual bit-rate $\mathcal{R}(\lfloor \bar{v}_t \rceil)$ (and thus the overall bit-rate) compared to STAT-SSF and SSF (with similar reconstruction quality), as illustrated in the third and fourth figures in the top row.

et al., 2020) in the shift transform, i.e., $h_\mu(\hat{x}_{t-1}, w_t) = h_{warp}(\hat{x}_{t-1}, g_w(w_t))$. This can also be seen as an extended version of the SSF model, whose shift transform $h_\mu$ is preceded by a new learned scale transform $h_\sigma$.

**Structured Prior (SP).** Besides improving the autoregressive transform (affecting the likelihood model for $x_t$), one variant of our approach also improves the topmost generative hierarchy in the form of a more expressive latent prior $p(z_{1:T})$, affecting the entropy model for compression. We observe that motion information encoded in $w_t$ can often be informative of the residual error encoded in $v_t$. In other words, large residual errors $v_t$ incurred by the mean prediction $h_\mu(\hat{x}_{t-1}, w_t)$ (e.g., the result of warping the previous frame $h_\mu(\hat{x}_{t-1})$) are often spatially collocated with (unpredictable) motion as encoded by $w_t$. The original SSF model's prior factorizes as $p(w_t, v_t) = p(w_t)p(v_t)$ and does not capture such correlation. We therefore propose a structured prior by introducing conditional dependence between $w_t$ and $v_t$, so that $p(w_t, v_t) = p(w_t)p(v_t|w_t)$. At a high level, this can be implemented by introducing a new neural network that maps $w_t$ to parameters of a parametric distribution of $p(v_t|w_t)$ (e.g., mean and variance of a diagonal Gaussian distribution). This results in variants of the above models, **STAT-SP** and **STAT-SSF-SP**, where the structured prior is applied on top of the proposed **STAT** and **STAT-SSF** models.

## 4 EXPERIMENTS

In this section, we train our models both on the existing dataset and our new YouTube-NT dataset. Our model also improves over state-of-the-art neural and classical video compression methods when evaluated on several publicly available benchmark datasets. Lower-level modules and training scheme for our models largely follow Agustsson et al. (2020); we provide detailed model diagrams and schematic implementation, including the proposed scaling transform and structured prior, in Appendix A.4. We also implement a more computationally efficient version of scale-space warping (Agustsson et al., 2020) based on Gaussian pyramid and interpolation (instead of naive Gaussian blurring of Agustsson et al. (2020)); pseudocode is available in Appendix A.3.

### 4.1 TRAINING DATASETS

**Vimeo-90k** (Xue et al., 2019) consists of 90,000 clips of 7 frames at 448x256 resolution collected from `vimeo.com`, which has been used in previous works (Lu et al., 2019; Yang et al., 2020a; Liu et al., 2020). While other publicly-available video datasets exist, they typically have lower

resolution and/or specialized content. e.g., Kinetics (Carreira & Zisserman, 2017) only contains human action videos, and previous methods that trained on Kinetics (Wu et al., 2018; Habibian et al., 2019; Golinski et al., 2020) generally report worse rate-distortion performance on diverse benchmarks (such as UVG, to be discussed below), compared to Agustsson et al. (2020) who trained on a significantly larger and higher-resolution dataset collected from `youtube.com`.

**YouTube-NT**. This is our new dataset. We collected 8,000 nature videos and movie/video-game trailers from `youtube.com` and processed them into 300k high-resolution (720p) clips, which we refer to as YouTube-NT. We release YouTube-NT in the form of customizable scripts to facilitate future compression research. Table 1 compares the current version of YouTube-NT with Vimeo-90k (Xue et al., 2019) and with Google's closed-access training dataset (Agustsson et al., 2020). Figure 5b shows the evaluation performance of the SSF model architecture after training on each dataset.

Table 1: **Overview of Training Datasets**.

| Dataset name | Clip length | Resolution | # of clips | # of videos | Public | Configurable |
|---|---|---|---|---|---|---|
| Vimeo-90k | 7 frames | 448x256 | 90,000 | 5,000 | ✓ | ✗ |
| YouTube-NT (**ours**) | 6-10 frames | 1280x720 | 300,000 | 8,000 | ✓ | ✓ |
| Agustsson 2020 et al. | 60 frames | 1280x720 | 700,000 | 700,000 | ✗ | ✗ |

## 4.2 TRAINING AND EVALUATION PROCEDURES

**Training**. All models are trained on three consecutive frames and batchsize 8, which are randomly selected from each clip, then randomly cropped to 256x256. We trained on MSE loss, following similar procedure to Agustsson et al. (2020) (see Appendix A.2 for details).

**Evaluation**. We evaluate compression performance on the widely used **UVG** (Mercat et al., 2020) and **MCL_JCV** (Wang et al., 2016) datasets, both consisting of raw videos in YUV420 format. UVG is widely used for testing the HEVC codec and contains seven 1080p videos at 120fps with smooth and mild motions or stable camera movements. MCL_JCV contains thirty 1080p videos at 30fps, which are generally more diverse, with a higher degree of motion and a more unstable camera.

We compute the bit rate (bits-per-pixel, BPP) and the reconstruction quality (measured in PSNR) averaged across all frames. We note that PSNR is a more challenging metric than MS-SSIM (Wang et al., 2003) for learned codecs (Lu et al., 2019; Agustsson et al., 2020; Habibian et al., 2019; Yang et al., 2020a;c). Since existing neural compress methods assume video input in RGB format (24bits/pixel), we follow this convention in our evaluations for meaningful comparisons. We note that HEVC also has special support for YUV420 (12bits/pixel), allowing it to exploit this more compact file format and effectively halve the input bitrate on our test videos (which were coded in YUV420 by default), giving it an advantage over all neural methods. Regardless, we report the performance of HEVC in YUV420 mode (in addition to the default RGB mode) for reference.

Table 2: **Overview of compression methods and the datasets trained on** (if applicable).

| Model Name | Category | Vimeo-90k | Youtube-NT | Remark |
|---|---|---|---|---|
| **STAT-SSF** | Proposed | ✓ | ✓ | Proposed autoregressive transform (on top of our efficient scale-space flow implementation) |
| **STAT-SSF-SP** | Proposed | ✓ | ✗ | Same as above (STAT-SSF) but combined with our proposed structured prior |
| SSF | Baseline | ✓ | ✓ | Agustsson et al. (2020) CVPR |
| DVC | Baseline | ✓ | ✗ | Lu et al. (2019) CVPR |
| VCII | Baseline | ✗ | ✗ | Wu et al. (2018) ECCV (trained on the Kinetics dataset) |
| DGVC | Baseline | ✓ | ✗ | Han et al. (2019) NeurIPS; modified for low-latency compression setup (no future frames used) |
| TAT | Baseline | ✓ | ✗ | Yang et al. (2020b) ICML workshop |
| HEVC | Baseline | N/A | N/A | State-of-the-art conventional codec with RGB 4:4:4 color space input |
| HEVC(YUV) | Baseline | N/A | N/A | State-of-the-art conventional codec with YUV 4:2:0 color space input |
| STAT | Ablation | ✓ | ✓ | Replace scale space flow in STAT-SSF with neural network |
| SSF-SP | Ablation | ✗ | ✓ | Scale space flow model with structured prior |

## 4.3 BASELINE ANALYSIS

We trained our models on Vimeo-90k to compare with the published results of baseline models listed in Table 2. Figure 3a compares our proposed models (STAT-SSF, STAT-SSF-SP) with previous state-of-the-art classical codec HEVC and neural codecs on the UVG test dataset. Our STAT-SSF-SP model provides superior performance at bitrates ≥ 0.07 BPP, outperforming conventional HEVC

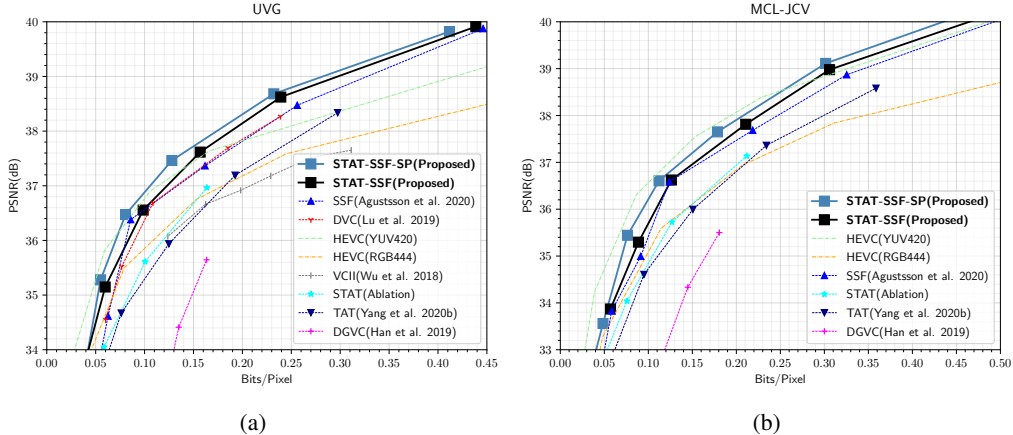

(a)                                                     (b)

Figure 3: **Rate-Distortion Performance** of various models and ablations. Results are evaluated on **(a)** UVG and **(b)** MCL_JCV datasets. All the learning-based models (except VCII (Wu et al., 2018)) are trained on Vimeo-90k. STAT-SSF-SP (proposed) achieves the best performance.

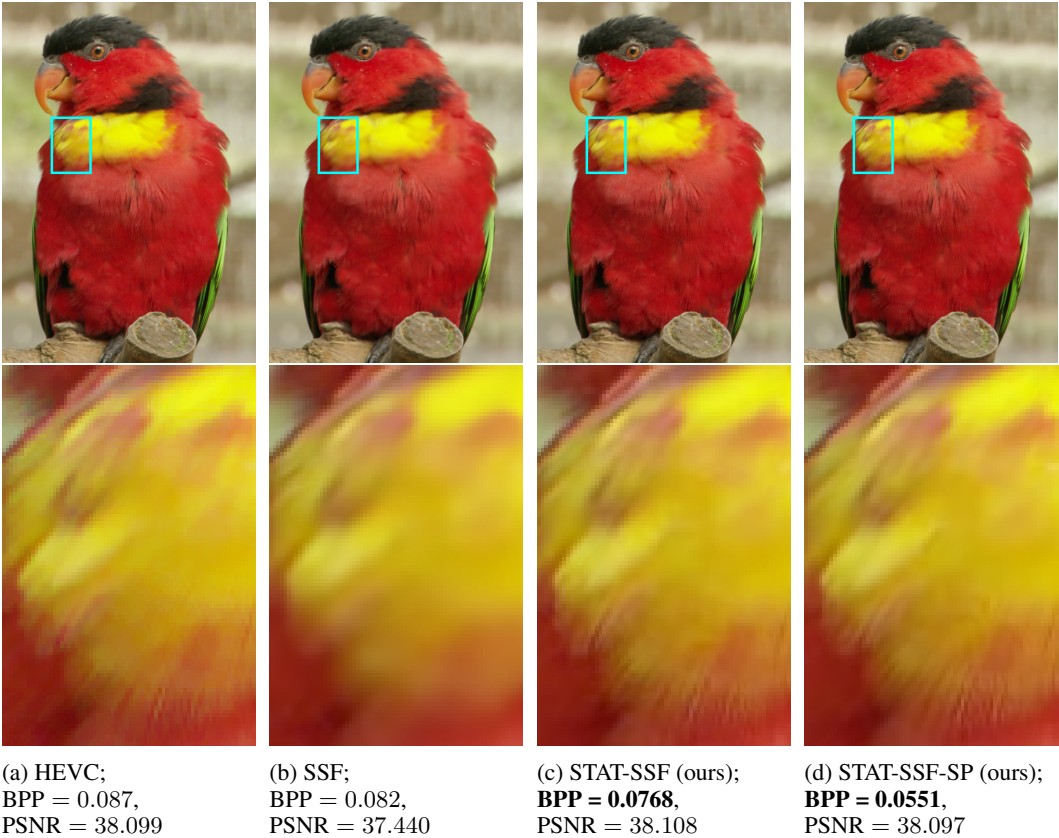

(a) HEVC;
BPP = 0.087,
PSNR = 38.099

(b) SSF;
BPP = 0.082,
PSNR = 37.440

(c) STAT-SSF (ours);
**BPP = 0.0768**,
PSNR = 38.108

(d) STAT-SSF-SP (ours);
**BPP = 0.0551**,
PSNR = 38.097

Figure 4: **Qualitative comparisons** of various methods on a frame from MCL-JCV video 30. Figures in the bottom row focus on the same image patch on top. Here, models with the proposed scale transform (STAT-SSF and STAT-SSF-SP) outperform the ones without, yielding visually more detailed reconstructions at lower rates; structured prior (STAT-SSF-SP) reduces the bit-rate further.

even in its favored YUV 420 mode and state-of-the-art neural method SSF (Agustsson et al., 2020), as well as the established DVC (Lu et al., 2019) model, which leverages a more complicated model and multi-stage training procedure. We also note that, as expected, our proposed STAT model improves over TAT (Yang et al., 2020b), with the latter lacking stochasticity in the autoregressive transform compared to our proposed STAT and its variants.

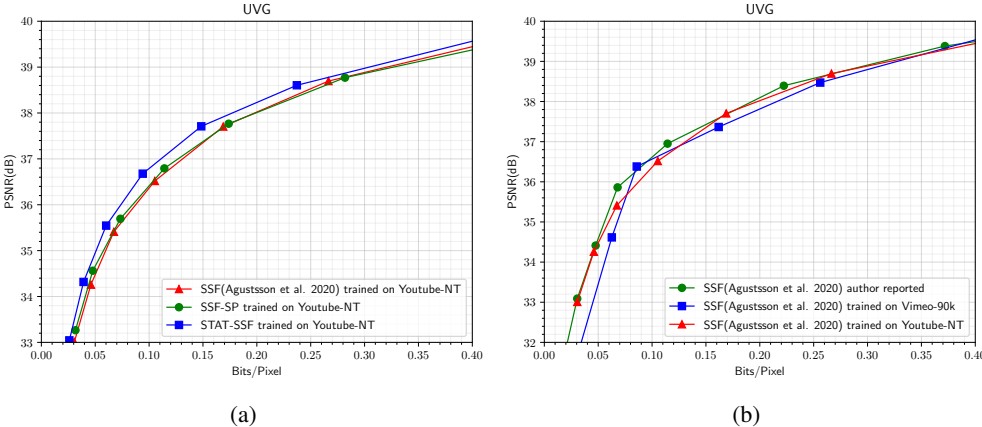

(a)                            (b)

Figure 5: **Ablations & Comparisons**. **(a)** An ablation study on our proposed components. **(b)** Performance of SSF (Agustsson et al., 2020) trained on different datasets. Both sets of results are evaluated on UVG.

Figure 3a shows that the performance ranking on MCL_JCV is similar to on UVG, despite MCL_JCV having more diverse and challenging (e.g., animated) content (Agustsson et al., 2020). We provide qualitative results in Figure 2 and 4, offering insight into the behavior of the proposed scaling transform and structured prior, as well as visual qualities of the top-performing methods.

## 4.4 ABLATION ANALYSIS

Using the baseline SSF (Agustsson et al., 2020) model, we study the performance contribution of each of our proposed components, stochastic temporal autoregressive transform (STAT) and structured prior (SP), in isolation. We trained on YouTube-NT and evaluated on UVG. As shown in Figure 5a, STAT improves performance to a greater degree than SP, while SP alone does not provide noticeable improvement over vanilla SSF (however, note that when combined with STAT, SP offers higher improvement over STAT alone, as shown by STAT-SSF-SP v.s. STAT-SSF in Figure 3a).

To quantify the effect of training data on performance, we compare the test performance (on UVG) of the SSF model trained on Vimeo-90k (Xue et al., 2019) and YouTube-NT. We also provide the reported results from Agustsson et al. (2020), which trained on a larger (and unreleased) dataset. As seen from the R-D curves in Figure 5b, training on YouTube-NT improves rate-distortion performance over Vimeo-90k, in many cases bridging the gap with the performance from the larger closed-access training dataset of Agustsson et al. (2020). At a higher bitrate, the model trained on Vimeo-90k(Xue et al., 2019) tends to have a similar performance to YouTube-NT. This is likely because YouTube-NT currently only covers 8000 videos, limiting the diversity of the short clips.

## 5 DISCUSSION

We provide a unifying perspective on sequential video compression and temporal autoregressive flows (Marino et al., 2020), and elucidate the relationship between the two in terms of their underlying generative hierarchy. From this perspective, we consider several video compression methods, particularly a state-of-the-art method Scale-Space-Flow (Agustsson et al., 2020), as sequential variational autoencoders that implement a more general stochastic temporal autoregressive transform, which allows us to naturally extend the Scale-Space-Flow model and obtain improved rate-distortion performance on standard public benchmark datasets. Further, we provide scripts to generate a new high-resolution video dataset, YouTube-NT, which is substantially larger than current publicly-available datasets. Together, we hope that this new perspective and dataset will drive further progress in the nascent yet highly impactful field of learned video compression.

## 6 ACKNOWLEDGEMENTS

We gratefully acknowledge extensive contributions from Yang Yang (Qualcomm), which were indispensable to this work. This material is based upon work supported by the Defense Advanced Research Projects Agency (DARPA) under Contract No. HR001120C0021. Any opinions, findings and conclusions or recommendations expressed in this material are those of the author(s) and do not necessarily reflect the views of the Defense Advanced Research Projects Agency (DARPA). Yibo Yang acknowledges funding from the Hasso Plattner Foundation. Furthermore, this work was supported by the National Science Foundation under Grants 1928718, 2003237 and 2007719, as well as Intel and Qualcomm.

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

## A APPENDIX

### A.1 COMMAND FOR HEVC CODEC

To avoid `FFmpeg` package taking the advantage of the input file color format (YUV420), we first need to dump the `video.yuv` file to a sequence of lossless `png` files:

```
ffmpeg -i video.yuv -vsync 0 video/%d.png
```

Then we use the default low-latency setting in `ffmpeg` to compress the dumped `png` sequences:

```
ffmpeg -y -i video/%d.png -c:v libx265 -preset medium \
    -x265-params bframes=0 -crf {crf} video.mkv
```

where `crf` is the parameter for quality control. The compressed video is encoded by HEVC with RGB color space.

To get the result of HEVC (YUV420), we directly execute:

```
ffmpeg -pix_fmt yuv420p -s 1920x1080 -i video.yuv \
    -c:v libx265 -crf {crf} -x265-params bframes=0 video.mkv
```

### A.2 TRAINING SCHEDULE

Training time is about four days on an NVIDIA Titan RTX. Similar to Agustsson et al. (2020), we use the Adam optimizer (Kingma & Ba, 2015), training the models for 1,050,000 steps. The initial learning rate of 1e-4 is decayed to 1e-5 after 900,000 steps, and we increase the crop size to 384x384 for the last 50,000 steps. All models are optimized using MSE loss.

### A.3 EFFICIENT SCALE-SPACE-FLOW IMPLEMENTATION

Agustsson et al. (2020) uses a simple implementation of scale-space flow by convolving the previous reconstructed frame $\hat{x}_{t-1}$ with a sequence of Gaussian kernel $\sigma^2 = \{0, \sigma_0^2, (2\sigma_0)^2, (4\sigma_0)^2, (8\sigma_0)^2, (16\sigma_0)^2\}$. However, this leads to a large kernel size when $\sigma$ is large, which can be computationally expensive. For example, a Gaussian kernel with $\sigma^2 = 256$ usually requires kernel size 97x97 to avoid artifact (usually $kernel\_size = (6 * \sigma + 1)^2$). To alleviate the problem, we leverage an efficient version of Gaussian scale-space by using Gaussian pyramid with upsampling. In our implementation, we use $\sigma^2 = \{0, \sigma_0^2, \sigma_0^2 + (2\sigma_0)^2, \sigma_0^2 + (2\sigma_0)^2 + (4\sigma_0)^2, \sigma_0^2 + $

$(2\sigma_0)^2 + (4\sigma_0)^2 + (8\sigma_0)^2, \sigma_0^2 + (2\sigma_0)^2 + (4\sigma_0)^2 + (8\sigma_0)^2 + (16\sigma_0)^2\}$, because by using Gaussian pyramid, we can always use Gaussian kernel with $\sigma = \sigma_0$ to consecutively blur and downsample the image to build a *pyramid*. At the final step, we only need to upsample all the downsampled images to the original size to approximate a scale-space 3D tensor. Detailed algorithm is described in Algorithm 1.

---

**Algorithm 1:** An efficient algorithm to build a scale-space 3D tensor

---

   **Result:** *ssv*: Scale-space 3D tensor
   **Input:** *input* input image; $\sigma_0$ base scale; $M$ scale depth;
   ssv = [input];
   kernel = Create_Gaussian_Kernel($\sigma_0$);
   **for** *i=0 to M-1* **do**
      input = GaussianBlur(input, kernel);
      **if** *i == 0* **then**
         ssv.append(input);
      **else**
         tmp = input;
         **for** *j=0 to i-1* **do**
            tmp = UpSample2x(tmp); {step upsampling for smooth interpolation};
         **end**
         ssv.append(tmp);
      **end**
      input = DownSample2x(input);
   **end**
   **return** Concat(ssv)

---

## A.4 LOWER-LEVEL ARCHITECTURE DIAGRAMS

Figure 6 illustrates the low-level encoder, decoder and hyper-en/decoder modules used in our proposed STAT-SSF and STAT-SSF-SP models, as well as in the baseline TAT and SSF models, based on Agustsson et al. (2020). Figure 7 shows the encoder-decoder flowchart for $\mathbf{w}_t$ and $\mathbf{v}_t$ separately, as well as their corresponding entropy models (priors), in the STAT-SSF-SP model.

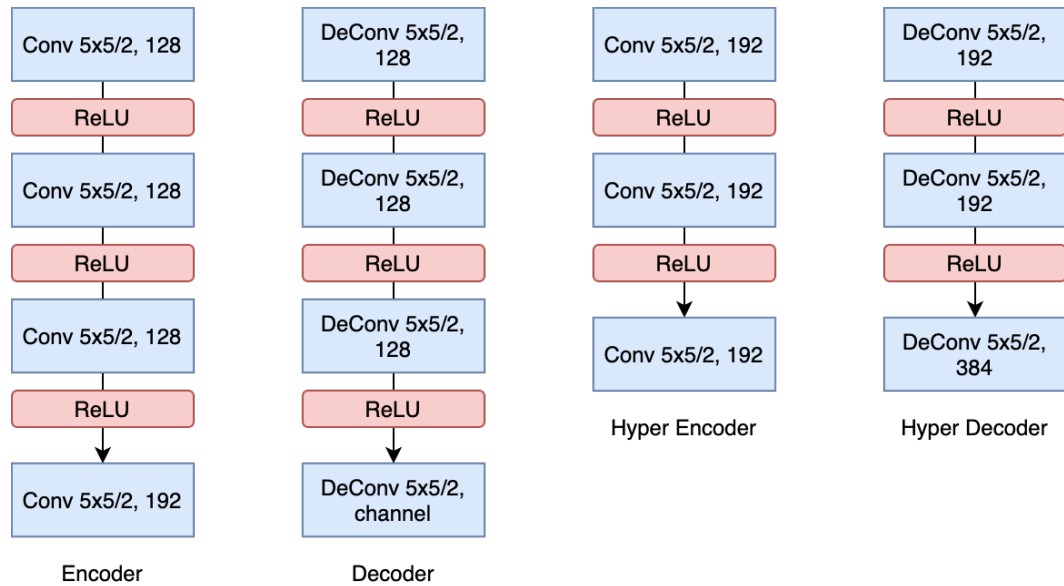

Figure 6: Backbone module architectures, where "5x5/2, 128" means 5x5 convolution kernel with stride 2; the number of filters is 128.

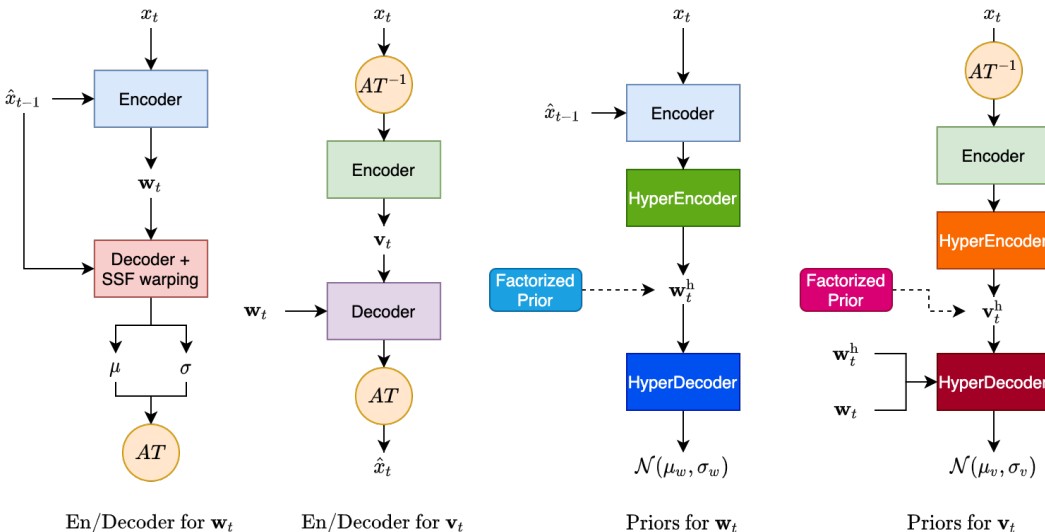

Figure 7: Computational flowchart for the proposed STAT-SSF-SP model. The left two subfigures show the decoder and encoder flowcharts for $\mathbf{w}_t$ and $\mathbf{v}_t$, respectively, with "AT" denoting autoregressive transform. The right two subfigures show the prior distributions that are used for entropy coding $\mathbf{w}_t$ and $\mathbf{v}_t$, respectively, with $p(\mathbf{w}_t, \mathbf{w}_t^{\mathrm{h}}) = p(\mathbf{w}_t^{\mathrm{h}})p(\mathbf{w}_t|\mathbf{w}_t^{\mathrm{h}})$, and $p(\mathbf{v}_t, \mathbf{v}_t^{\mathrm{h}}|\mathbf{w}_t, \mathbf{w}_t^{\mathrm{h}}) = p(\mathbf{v}_t^{\mathrm{h}})p(\mathbf{v}_t|\mathbf{v}_t^{\mathrm{h}}, \mathbf{w}_t, \mathbf{w}_t^{\mathrm{h}})$, with $\mathbf{w}_t^{\mathrm{h}}$ and $\mathbf{v}_t^{\mathrm{h}}$ denoting hyper latents (see (Agustsson et al., 2020) for a description of hyper-priors); note that the priors in the SSF and STAT-SSF models (without the proposed structured prior) correspond to the special case where the HyperDecoder for $\mathbf{v}_t$ does not receive $\mathbf{w}_t^{\mathrm{h}}$ and $\mathbf{w}_t$ as inputs, so that the entropy model for $\mathbf{v}_t$ is independent of $\mathbf{w}_t$: $p(\mathbf{v}_t, \mathbf{v}_t^{\mathrm{h}}) = p(\mathbf{v}_t^{\mathrm{h}})p(\mathbf{v}_t|\mathbf{v}_t^{\mathrm{h}})$.

