# OpenReview forum: "Hierarchical Autoregressive Modeling for Neural Video Compression"
_ICLR.cc/2021/Conference — ICLR 2021 Poster_

### Official Review · AnonReviewer2 · 2020-10-21
**This work brings a new interpretation of learned video compression and shows promising performance.**

**Rating:** 7
**Confidence:** 3

**Review:**

#### Summary
In this paper, the authors provide a new interpretation of existing video compression models. Their perspective is that a video decoder is a stochastic temporal autoregressive model with latent variables. The introduced latent variables could be either used for providing more expressive power for 1) motion estimation&compensation modeling and 2) residual noise modeling, which are two key components of traditional video codecs. The proposed method shows favorable results when the bitrate is higher than 0.12 bits per pixel on the public benchmarks.

#### Strength
The proposed method has the following strengths.
* The new interpretation of the video compression model gives a unified framework and insights. Although I am not entirely familiar with the learned video compression field, the interpretation looks quite interesting. It would be helpful for other researchers in the field.
* New moderately high-resolution dataset is provided for training learned video compression models. This is a good contribution to the community.
* The proposed model improves performance over the baselines. They propose two new components: 1) a learnable scaling transform for modeling motion estimation/compensation, and 2) structured priors for modeling the entropy.

To make the work stronger, I have a few suggestions as follows.
* Study on the effect of using a pre-trained optical flow model. It seems that the parameters for motion estimation (g_w, f_w, w_t) are learned end-to-end. However, it might be challenging to learn motion estimation parameters end-to-end with other parameters. I am curious how much gain we can get if we plug in the pre-trained optical flow estimator. Intuitively it should give some boost, as one major component of the video codec is the motion estimator.
* More failure mode analyses. The proposed models, STAT and STAT-SSF, show favorable results when the bitrate is high but underperforms baselines in low bitrate. It would be better to have some discussion about why the case is.
* High latency extension. It would be interesting to see how the proposed framework can be extended for high-latency compression. In traditional video codecs, high latency compressions are also widely used to achieve high rate-distortion performance. However, in this work, only the frame t and t-1 are considered for compression.


#### Minor comments
* I am not sure the authors can cite papers in the abstract.

---

> ### Author Response · Authors · 2020-11-17
> **Response to AnonReviewer2**
>
> > Study on the effect of using a pre-trained optical flow model. It seems that the parameters for motion estimation (g_w, f_w, w_t) are learned end-to-end. However, it might be challenging to learn motion estimation parameters end-to-end with other parameters. I am curious how much gain we can get if we plug in the pre-trained optical flow estimator. Intuitively it should give some boost, as one major component of the video codec is the motion estimator.
>
> A pre-trained optical flow module is indeed a reasonable option, and has been used in previous work such as DVC (Lu et al. 2019) and HLVC (Yang et al. 2020a). However, recent results from SSF (Agustsson et al. 2020) indicate that an optical flow module trained as part of the larger neural compression model actually offers superior compression performance, compared to methods based on pre-trained optical flow modules. There are also practical reasons to favor an end-to-end approach to motion estimation for compression, which are outlined in the Introduction section of the SSF paper, and we summarize the main points as follows: 1). a pre-trained optical flow module targets a harder and more general motion estimation problem which is not aligned with the rate-distortion objective of video compression, while imposing unnecessary constraints to a neural compression model; 2). training an optical flow module itself can be expensive, requiring annotated flow data and complicating the overall training procedure.
>
> From our experiments with SSF-based models, we generally have little difficulty optimizing the entire model end-to-end, and visual inspection of the resulting optical flow module indicates it has learned very reasonable estimates of motion information, similar to that learned by the optical flow module in vanilla SSF. Thus we have reasons to believe that the findings in the SSF paper would carry over to our proposed models (which are built on SSF), and that a pre-trained optical flow module is unlikely to improve our performance.
>
> > More failure mode analyses. The proposed models, STAT and STAT-SSF, show favorable results when the bitrate is high but underperforms baselines in low bitrate. It would be better to have some discussion about why the case is.
>
> Please see our general comment #2 about suboptimal performance at lower bit-rates, and how we have addressed this in our updated Figure 3.
>
> > High latency extension. It would be interesting to see how the proposed framework can be extended for high-latency compression. In traditional video codecs, high latency compressions are also widely used to achieve high rate-distortion performance. However, in this work, only the frame t and t-1 are considered for compression.
>
> Indeed, there should be more performance gain from a high-latency setup (at the cost of higher latency, of course). Generally, this can be achieved by using a larger temporal receptive field for motion estimation/compensation as you suggested, e.g., by including more than one previous frame (which is straightforward to do in our model), or even including future frames (e.g., frame t+1, etc.) as considered by frame interpolation methods (Wu et al., 2018), allowing for more expressive conditioning in the generative/inference process, and/or incorporating more "global" latent variables that encode information on a longer time scale (Han et al., 2019).
>
> Since our work builds on SSF and compares with other existing state-of-the-art approaches in the low-latency setting, we leave high-latency extensions to future work.

---

### Official Review · AnonReviewer3 · 2020-10-28
**Regarding major contributions and experiments**

**Rating:** 6
**Confidence:** 4

**Review:**

The paper presents a lossy video coding scheme using autoregressive generative models.
+ It is good to see that the paper includes comprehensive reviews regarding neural network based compression schemes and autoregressive models.

- The contribution has been depicted in aspects of 1) a new framework, 2) a new model, and 3) a new dataset. However, beside to 3), it is hard to see the other two aspects have significant novelty both in a neural video coding architecture (Agustsson et al. 2020) and in temporal autoregressive models, etc.

- Furthermore, current video codecs (i.e. HEVC) are developed for YUV4:2:0 because the current broadcasting system has supported the video format. The evaluation should be conducted in YUV 4:2:0 domain for fair and reasonable comparisons.

---

> ### Author Response · Authors · 2020-11-17
> **Response to AnonReviewer3**
>
> > The contribution has been depicted in aspects of 1) a new framework, 2) a new model, and 3) a new dataset. However, besides 3), it is hard to see the other two aspects have significant novelty both in a neural video coding architecture (Agustsson et al. 2020) and in temporal autoregressive models, etc.
>
> Admittedly, the connection between generative video modeling and sequential VAE is not new (Han et al., 2019); however we are the first to draw connections to normalizing flow literature in the field of video compression that is often dominated by engineering concerns, and also the first to establish connections between many existing video compression codecs and a type of hybrid VAE-autoregressive-flow model in deep generative modeling (Marino et al., 2020),  formalizing the underlying probabilistic generative processes of established neural compression methods such as SSF and DVC. As we remarked in our general comment #2, this generative modeling framework naturally leads us to consider extensions of SSF (Agustsson et al. 2020), which we believe are valuable contributions on their own, and also promotes knowledge transfer between the more established generative modeling literature and the developing field of neural video compression.
>
> > Furthermore, current video codecs (i.e. HEVC) are developed for YUV4:2:0 because the current broadcasting system has supported the video format. The evaluation should be conducted in YUV 4:2:0 domain for fair and reasonable comparisons.
>
> As per your suggestion, we added results for HEVC in YUV420 mode in our latest rate-distortion curves (see Figure 3), and our proposed methods still remain competitive, outperforming HEVC YUV420 for bit-rate >= 0.07 on UVG, and for bit-rate >=0.25 on MCL_JCV (in addition to outperforming all neural baselines). However, as we pointed out in Section 4.2 of the paper, evaluating HEVC in YUV420 mode (instead of RGB) is not very meaningful for assessing our contribution to neural video compression.
>
> YUV420 encoding provides a more compact video representation than RGB (due to chroma subsampling), using 12 bits per pixel instead of RGB’s 24 bits per pixel, and this is the default file format of all the test videos (UVG and MCL_JCV) considered in this paper. However, existing neural compression methods (e.g., DVC, SSF) do not exploit this fact, and instead are trained on videos in the more widely available RGB format. We, therefore, follow this convention to better compare with existing methods. Evaluating HEVC in YUV420 mode would allow HEVC to exploit the more compact YUV420 file format and effectively halve the bit rate of the source videos, which 1). is only possible for the UVG and MCL_JCV videos considered here, and does not work for other test videos with more precise chroma sampling (e.g. 422 or 444) while RGB can losslessly represent all these formats, and 2). gives HEVC an advantage over all existing neural compression methods, which we believe is irrelevant for assessing our proposed improvements to neural video compression.

---

### Official Review · AnonReviewer4 · 2020-10-28

**Rating:** 7
**Confidence:** 4

**Review:**

This paper proposes improved version of Scale-Space Flow (SSF) model. Authors add two enhancements to SSF for improve Rate-Distortion performance: (i) learnable scale transform and (ii) structured prior (SP). The paper also introduce a dataset for neural video compression that collected from youtube.com.

Pros:
- Proposal seems better results than conventional SSF in some bit-rate and dataset conditions (Figure 3).
- Publishing a new dataset may help the community.

Cons:
- I think the introduction of scale parameter is the most important point of this paper. Authors says it acts as a gating mechanism. Figure 2 shows the case of an image, but no comparison with traditional SSF has been made. Figure 4 shows final RD-curve comparison from traditional SSF but I don't know how the gating mechanism works. For example, showing some kind of residual noise amount of conventional SSF (Agustsson et al., 2020) and show a reduce effect by scale parameter will make the claim of the paper credible.
- There needs to be a more specific and clear explanation of how the SP is processed.
- Figure 4 shows the effect of SP is that the image quality is better than SSF at high bitrates, but the image quality is worse at low bitrates. I would like to see a discussion on the reason for this.

---

> ### Author Response · Authors · 2020-11-17
> **Response to AnonReviewer4**
>
> > I think the introduction of the scale parameter is the most important point of this paper.
>
> In our initial results, the proposed scale parameter did make the most empirical difference. However, as we detailed in general comment #3, our contribution is not limited to proposing the scaling transform. Moreover, we were able to improve the performance of structured prior after further experiments (see updated Figure 3, and general comment #4), making it a significant contribution of its own.
>
> > Authors say it acts as a gating mechanism. Figure 2 shows the case of an image, but no comparison with traditional SSF has been made. Figure 4 shows the final RD-curve comparison from traditional SSF but I don't know how the gating mechanism works. For example, showing some kind of residual noise amount of conventional SSF (Agustsson et al., 2020) and show a reduced effect by scale parameter will make the claim of the paper credible.
>
> We very much appreciate your constructive feedback.  The newly included Figure 4 compares the reconstruction quality of our proposed models (STAT-SSF, STAT-SSF-SP) against top baseline methods (HEVC and SSF) side-by-side, at equal or lower bit-rate.
> And as per your suggestion, we’ve updated Figure 2 to show heatmaps of residual bitrate savings of STAT-SSF-SP relative to ablated variants STAT-SSF (without the proposed structured prior) and SSF (without the proposed structured prior or scaling transform), at comparable reconstruction quality (as measured in PSNR). At a high level, the idea is that the proposed scale parameter $\sigma$ gives the model extra flexibility when combining the noise $y$ (decoded from ($v$, $w$)) with the warping prediction $\mu$ (decoded from $w$ only) to form the reconstruction $\hat x = \mu +\sigma * y$. In our example, the scale $\sigma$ down-weights contribution from the noise $y$ in the foreground where it is very costly, and reduces the residual bit-rate R($v$) (which dominates the overall bit-rate) when compared to STAT-SSF and SSF.
>
> > There needs to be a more specific and clear explanation of how the SP is processed.
>
> Following your suggestion, we have provided more details about the computation of the structure prior to our updated draft.
> Again, the idea of a structured prior is motivated by the observation that the motion information (encoded in $w$) can often be informative of the residual information (encoded in $v$), with the two being spatially correlated, thus encoding $v$ conditioned on w (via a conditional prior $p(v|w)$) results in a better residual entropy model (and thus lower residual bit rate). As we describe in Section 3, this conditioning can be implemented by introducing a convolutional neural network that maps $w$ to parameters characterizing the distribution of $p(v|w)$ (e.g., mean and covariance of a diagonal Gaussian distribution, which we used in this work). At the beginning of Section 4, we also refer readers to Appendix A.4. for details of the lower-level encoder/decoder modules used (e.g., convolution/deconvolutional networks with hyper-parameters such as the number of layers and kernel size) as well as computational diagrams specifying how the structured prior is implemented.
>
> > Figure 4 shows the effect of SP is that the image quality is better than SSF at high bitrates, but the image quality is worse at low bitrates. I would like to see a discussion on the reason for this.
>
> First, we note that Figure 4 (now Figure 5 in the updated draft) is only a side result of our ablated model. Structured prior improved compression performance of our main proposed model (STAT-SSF) across *all* bitrates in Figure 3.
> Second, as we mentioned in general comment #2, models trained at low bitrates may present slightly suboptimal performance, and strategies like beta-annealing can help. We do not yet have updated results for the ablation models in question, as training high-performing video compression models are quite time and resource-intensive, but hope to be able to update on this soon.

---

> > ### Comment · AnonReviewer4 · 2020-11-21
> > **Thanks**
> >
> > Thank you very much for updating and explaining the paper. My concerns have been dispelled.
> > I have updated my rating.

---

### Official Review · AnonReviewer1 · 2020-10-29
**Interesting Application of Contemporary Generative Models**

**Rating:** 7
**Confidence:** 3

**Review:**

In this paper, the authors focus on the problem of lossy video compression. To this end they propose the application of latent variable sequential generative models, specifically autoregressive flows to compress video streams. They evaluate variations of these models quantitatively including their own proposed version of scale space flow. They also introduce a new dataset named Youtube-NT and show promising quantitative performance.

Pros:

1. The proposed problem of video compression has direct societal applications.

2. The application of generative models, especially flow based models, to video compression is a novel and relatively underexplored topic in the community.

3. Quantitative performance is promising.

4. New dataset released will be useful for future work.

Cons:

1. There ideally should be more qualitative evaluation. How do the reconstructions of all these approaches look side by side?

2. The new dataset is fairly small by the standards of video datasets. For example, Kinetics and Youtube 8m have hundreds of thousands or even millions of examples.

In summary, the paper explores an interesting and novel application of flow based models to video compression. While I think the paper could be strengthened by more qualitative examples and a larger dataset, I think this could be a good contribution to the conference.

---

> ### Author Response · Authors · 2020-11-17
> **Response to AnonReviewer1**
>
>
> > There ideally should be more qualitative evaluation. How do the reconstructions of all these approaches look side by side?
>
> Thank you for the suggestion; we have implemented it by including new qualitative evaluation results in the updated draft (see general comment #1). In Figure 4, we compare the reconstruction quality of our proposed models (STAT-SSF, STAT-SSF-SP) against the top baseline methods (HEVC and SSF) side-by-side, at equal or lower bit-rate. Figure 2 plots heatmaps of residual bit-rate savings of STAT-SSF-SP relative to its ablated variants, STAT-SSF (without the proposed structured prior) and SSF (without the proposed structured prior or scaling transform), at comparable reconstruction quality (PSNR). The heatmaps show progressively larger rate savings going from STAT-SSF to STAT-SSF-SP, and from SSF to STAT-SSF to STAT-SSF-SP.
>
> > The new dataset is fairly small by the standards of video datasets. For example, Kinetics and Youtube 8m have hundreds of thousands or even millions of examples.
>
> It is true that our dataset has a smaller number of clips (300K) compared to Kinetics (650K), Agustsson et al.’s proprietary data set (700K), or YouTube-8m (8M), but this is a misleading metric for evaluating a dataset for neural video compression. Training state-of-the-art video compression models require high-resolution videos ideally without the compression artifacts of existing lossy video codecs such as MPEG-4. As we pointed out in Section 4.1, existing video datasets like Kinetics and Youtube-8m do not meet this criterion, and are often limited in terms of content diversity (e.g., Kinetics only contains videos with human actions, intended for human action recognition).
>
> By contrast, for our dataset (YouTube-NT) we collected only high-quality Youtube videos with at least 1080p resolution and downsampled them to 720p to remove compression artifacts. These videos from nature documentary and movie trailer channels offer high content diversity, and the common training strategy of using small (e.g., 256x256) random image crops (Agustsson et al. 2020,  Habibian et al. 2019) further augment the data diversity.
>
> As additional evidence for “quality over quantity” of training data, we note that CLIC (CVPR Workshop and Challenge on Learned Image Compression) provides training data of only around 1600 high-resolution images, which is significantly smaller than say ImageNet but is widely used for training state-of-the-art neural image compression models (Yang et al., 2020d).

---

### Author Response · Authors · 2020-11-17
**General Comments**

We thank all reviewers for their helpful feedback and suggestions, which we have fully incorporated in our updated paper draft.  Below we address common concerns shared by multiple reviewers, before responding to each reviewer individually.

1. Adding more qualitative evaluations (reviewers 1 and 4): We updated Figure 2 to offer more insight into the behavior of the proposed scaling transform and structured prior. A new Figure 4 has also been added to offer side-by-side visual comparisons of reconstruction qualities of our method against state-of-the-art conventional and neural methods, showing qualitative improvements from our approach.
2. Analyzing and addressing the lower rate-distortion performance in the low bit-rate regime (reviewers 2 and 4):
Based on our empirical results, training SSF-based models can encounter optimization issues and converge to sub-optimal models at lower bit-rates. We found a beta-annealing approach helpful for converging to better solutions at low bit-rate, where we initialize to a well-trained model with a higher bit-rate (thus smaller beta hyperparameter), then gradually increase beta throughout training until the target beta is reached. We were able to improve the rate-distortion results in Figure 3 with this fine-tuning procedure. More generally, neural compression methods can give suboptimal or unstable results at low bit-rates due to the rounding approximation used in training (Ballé et al., 2020, Figures 3 and 8).
3. Clarifying our contributions (reviewers 3 and 4) :

      **a.** We presented a unifying framework for sequence compression with VAEs and autoregressive flows/transform: We are the first to draw connections to normalizing flow literature in the field of video compression often dominated by engineering concerns, and also the first to establish connections between many existing video compression codecs and a type of hybrid VAE-autoregressive-flow model in deep generative modeling (Marino et al., 2020), formalizing the underlying probabilistic generative processes of established neural compression methods such as SSF and DVC. This naturally led us to the following extensions to the SSF model, which would have been less obvious without this framework.

      **b.** We introduced a new scaling transform to allow for a more expressive reconstruction transform (which has precedence in normalizing flow literature), and a new structured prior to jointly and more efficiently encode the motion and residual information (which we believe is novel in neural video compression). These two extensions target different and complementary aspects of a neural compression model (decoder/encoder, entropy model), and improve the compression performance over state-of-the-art conventional and neural baselines when evaluated on several public test datasets.

      **c.** We provided a new dataset, based on which we obtained similar compression results to those reported by (Agustsson et al. 2020) (which trained on a larger proprietary dataset).

    Accordingly, we revised and extended the list of contributions at the end of the Introduction section, particularly highlighting the two aspects of our newly proposed model (more expressive reconstruction through scaling transform and structured prior/entropy model) in relation to our overall generative modeling framework.

Besides the changes mentioned above, we made the following significant updates to the draft:

4. We updated Figure 3 with improved rate-distortion curves for both the SSF baseline and our proposed methods after further experiments with beta-annealing (see general comment #2) and following the training procedure of (Agustsson et al. 2020) more closely.
We also included the performance of HEVC in YUV420 mode for completeness (at R3’s request). We outperform HEVC (YUV420) on medium and high bitrates (when bit-rate >= 0.07 on UVG, and bit-rate >=0.25 on MCL_JCV), as well as all other baselines as before.
5. We updated Fig 1. (c) to highlight both aspects of our newly proposed model (also see above comment #3).
6. We provide more motivation and explanation for the structured prior in Section 3, and clearly refer to relevant parts of the Appendix for further implementation details.

References:

* Agustsson et al., 2020. Scale-space flow for end-to-end optimized video compression. CVPR 2020.
* Ballé et al., 2020. Nonlinear Transform Coding. arXiv:2007.03034.
* Lu et al., 2019. DVC: An End-to-end Deep Video Compression Framework. CVPR 2019.
* Han et al., 2019. Deep Generative Video Compression. NeurIPS 2019.
* Wu et al., 2018. Video Compression through Image Interpolation. ECCV 2018.
* Yang et al., 2020a. Learning for Video Compression with Hierarchical Quality and Recurrent Enhancement. CVPR 2020.
* Yang et al., 2020d. Improving Inference for Neural Image Compression. NeurIPS 2020.

---

### Decision · Program_Chairs · 2021-01-07
**Final Decision**

**Decision:**

Accept (Poster)

**Comment:**

All reviewers recommend acceptance. The authors have addressed several of the reviewers' concerns in their comments, conducted additional experiments, and updated the manuscript accordingly.

A concern was raised regarding the size of the dataset introduced and used by the authors for this work. However, I agree with the authors that it doesn't necessarily make sense to compare this to datasets designed for training video classification and/or generation models; In the compression setting, the quality of individual data points matters much more than their quantity, as the authors argue.

Reviewer 2 was curious about the potential of a pre-trained optical flow module. I believe the authors have convincingly argued that end-to-end learning is likely to be more effective and practical (and indeed, there is plenty of evidence for this in other ML contexts where training data is not scarce). I agree that a direct comparison in the paper would have been interesting, but this would constitute a significant investment of time and effort on the authors' part (as they also point out, training such a module separately could actually be more difficult), and I think it would be unreasonable to make this a condition for acceptance.